# The Systematic Workplace-Improvement Needs Generation (SWING): Verifying a Worker-Centred Tool for Identifying Necessary Workplace Improvements in a Nursing Home in Japan

**DOI:** 10.3390/ijerph19031671

**Published:** 2022-02-01

**Authors:** Tomoo Hidaka, Sei Sato, Shota Endo, Hideaki Kasuga, Yusuke Masuishi, Takeyasu Kakamu, Tetsuhito Fukushima

**Affiliations:** Department of Hygiene and Preventive Medicine, Fukushima Medical University, Fukushima 960-1295, Japan; marubun_sei@yahoo.co.jp (S.S.); shota-e@fmu.ac.jp (S.E.); h-kasuga@fmu.ac.jp (H.K.); masuishi@fmu.ac.jp (Y.M.); bamboo@fmu.ac.jp (T.K.); t-fuku@fmu.ac.jp (T.F.)

**Keywords:** work improvement, workplace improvement, worker-reported outcome, Schedule for the Evaluation of Individual Quality of Life, psycho-social factor, social determinants of health, quality of work life, occupational health management, work engagement

## Abstract

This study developed and tested a new measurement instrument, the Systematic Workplace-Improvement Needs Generation (SWING), to identify workplace-improvement needs. The participants were 53 workers in a Japanese nursing home for the elderly. The respondents used the SWING questionnaire to self-generate five ‘cues’ they considered important to improve the workplace. The workers determined each cue’s sufficiency level and weight balance (importance), and then we summarised the 265 cues into 21 categories for workplace improvements. The respondents identified the following items as the most important and the least sufficiently provided areas for workplace improvement: ‘interaction with customers’, ‘physical and psychological harassment’, ‘rewarding and challenging work’, and ‘sharing goals and objectives’. Although the workplace-improvement recommendations differed greatly from person to person, SWING prioritised the items by weight (importance) and sufficiency (current status), allowing organisations to address the needed improvements systematically. The SWING tool effectively elicited and prioritised respondents’ recommendations for improving the workplace. Because its items are self-generated by the respondents, SWING can be used for any occupation or workplace. Visualisation with bubble plots to clarify the improvement needs is incorporated into SWING.

## 1. Introduction

### 1.1. Background

In June 1999, the International Labour Organization (ILO) proposed its Decent Work agenda to ‘promote opportunities for women and men to obtain decent and productive work, in conditions of freedom, equity, security and human dignity’ [1]. It described universal rights related to working conditions, work environments, and occupational health management.

The psychological scales created in previous studies have been used for quantifying decent work by measuring job satisfaction [2,3], perceived support [4], job stress and its responses such as burnout [5,6,7,8,9,10,11], interpersonal relationships [12], and quality of work life [13], as exemplified in Table 1.

The Minnesota Job Satisfaction Questionnaire [2], published in 1967, is a classic scale related to work improvement, and is widely used today. For example, in surveys of job satisfaction among healthcare workers [15], it is useful because it can collect a wide variety of information. Job satisfaction is also explored in the Job Satisfaction Survey (JSS) [3]. The JSS was developed to explore job satisfaction among workers in the public and welfare sectors who were not adequately recognised compared to those in the industrial sector. Thus, the measurement of job satisfaction has been expanded to cover a wide variety of workers.

The Multidimensional Scale of Perceived Social Support (MSPSS) [4] is characterised by its focus on examining a worker’s social support rather than the worker’s own traits. Because of its unique feature of measuring interpersonal relationships, the MSPSS is often used to identify mediating or confounding factors for the psychological burden of work and health problems as a response to such a burden [16].

Many scales have been developed to investigate job stress and burnout. The Maslach Burnout Inventory-General Survey (MBI-GS) [5], a key measure of burnout, defines burnout as three factors: exhaustion, cynicism, and professional efficacy. Because burnout is closely related to factors such as workload and worker personality, the MBI-GS is often used in conjunction with scales/instruments that measure these factors. The Job Content Questionnaire [6] examines the nature of work with questions such as ‘My job requires working very fast’. The Brief Job Stress Questionnaire (BJSQ) [7,8,9] is a scale that covers all aspects of physical, psychological, and social health, and includes items related to positive aspects of work, such as work engagement [17]. The Copenhagen Psychosocial Questionnaire [10] is a scale designed for practical interventions to improve the workplace, by measuring psychosocial work-related factors from three dimensions: workplace, work-individual interface, and individual. The Mental Health Improvement & Reinforcement Research of Recognition [11] is a mental health-specific instrument developed to examine the relationship between work conditions and their burden in detail. Significant efforts have been made to include items in these scales that measure the worker’s situation as comprehensively as possible.

The Perceptions of Fair Interpersonal Treatment scale [12], a psychological measure of interpersonal treatment, is an example of a scale that focuses on particular aspects rather than the overall psychological burden of work. This scale has been used in research on fairness and justice in workplaces, such as the gender disparities in disrespectful behaviour in the workplace [18].

The Quality of Worklife Questionnaire [13], developed by the National Institute for Occupational Safety and Health in the U.S., consists of items related to management, such as working hours, and organisational issues, such as corporate culture. The scale has been used in longitudinal studies spanning several decades, and these studies have provided basic information about workers [19,20]. As this scale also includes items related to workers’ occupational safety, it is sometimes used in safety research, such as injury prevention studies [21].

### 1.2. Assumptions and Limitations of Previous Studies

These above-mentioned previous studies aimed to collect information related to work and the workplace that could be used to improve work, by self-reports from the subjects. These previous studies have in common the methodological fact that the subjects answered questionnaire items that the researchers had judged to be important and were thus employed in their respective scales. Such procedures for generating question items and the measurements using them have been used as a matter of course in psychological scales. However, these procedures and measurements may have the following methodological limitations.

First, most psychological scales and questionnaires for workplace improvement fail to capture the unique problems of individual workplaces and workers because they use general items that are broadly applicable to a variety of workplaces [22]. Not all items researchers assume to be important for workplace improvements are necessary for all workers in all companies. Previous instruments have not incorporated ways to generate and evaluate items from the workers’ viewpoints inductively.

Second, few previous instruments have used assessment methods that allow workers to rate the relative importance of their work-improvement needs; the degree of sufficiency and subjective weighting for questionnaire items are different concepts that should be distinguished. As shown in Table 1, many questionnaires use the Likert scale measurement method, but do not distinguish between sufficiency and weight, which would not deepen our contextual understanding of workers’ concerns.

Consequently, there is no comprehensive instrument for subjectively identifying workers’ needs for workplace improvement. To overcome the methodological limitations mentioned above, that is, inductive generation of items appropriate for individual workplaces from the perspective of workers and distinction between sufficiency and weight, it is rational to fundamentally change the measurement procedures. We responded to this demand by proposing a new measurement method: the systematic workplace-improvement needs generation (SWING).

### 1.3. SWING’s Specifications

#### 1.3.1. Procedures and Advantages

Appendix A show Japanese and English versions of an example of a self-administered SWING questionnaire asking respondents to do the following: (1) describe five ‘cues’ for the question ‘What kind of workplace is easy/comfortable for you to work in?’ to subjectively generate items reflecting the workers’ work-improvement needs; (2) answer the question ‘To what extent are the five things you just listed currently fulfilled?’ and rate their sufficiency level on a scale of 0–100; and (3) answer the question ‘What is the balance of importance of the five items?’ to weight balance the relative importance of the cues. We then multiplied the sufficiency level and weight values (%) for each cue, summing the resulting values to obtain an index score. The index score (0–100) represented how satisfied the workers were with their work or workplace, with a higher number indicating a higher satisfaction level.

For (1) item generation, data are provided by the subject’s free description. The content of such descriptions may be brief or redundant. Accordingly, the researcher needs to group these descriptions inductively/qualitatively. Because of the nature of qualitative analysis, the SWING method will yield different workplace-improvement needs items depending on the workplace where the survey was conducted, or more precisely, the workers who cooperated in the survey. Given that this trait, which results from ‘cues’ and consequent ‘items’ in the SWING method that vary depending on the workplace, consistency or reproductivity of such ‘cues’ and ‘items’ is not supported by a (large) sample size of subjects, nor by a criterion of reliability such as test–retest reliability, where the quality of the measurement is ensured by the similarity of the results between surveys. Instead, the appropriateness of SWING as a method should be explored in terms of its data collection procedures: an inductive generation of items that would contribute to the practice of workplace improvement. SWING can be used by any organisation regardless of the number of employees and provides ideas for workplace improvements that are considered to be best for individual workplaces.

The distinction between (2) sufficiency and (3) weight is a feature of the quantification procedure in SWING. The importance of this distinction will be illustrated in the following example of the item of the BJSQ: ‘I can reflect my opinions on workplace policy’. If a subject responds ‘not at all’ to this item, he/she are reporting a low level of sufficiency. In contrast, they may perceive that their own participation in the workplace policy has little worth or less interest; in this case, the weight may be low. Past studies did not distinguish between sufficiency and weight, although items with lower weights may have lower priority as workplace-improvement needs. SWING has the advantage of identifying truly high-priority workplace-improvement needs in individual workplaces by first extracting specific needs for workplace improvement by inductive item generation and then evaluating them quantitatively by distinguishing between sufficiency and weight details.

#### 1.3.2. SWING’s Expected Contributions to Occupational Health, Theoretical Appropriateness, and Originality

SWING identifies workplace-improvement needs from the workers’ perspective, providing valuable information for establishing a participatory work improvement that promotes occupational safety and health and comprehensive risk management for both workers and employers [23]. Thus, we believe that the rich data provided by SWING can help promote occupational safety, health, and management.

SWING’s worker-centred procedure elicits qualitative data (e.g., the five cues) and quantitative data (e.g., sufficiency level, weight balance, and index score). Therefore, it was difficult to compare its validity and reliability as a psychological scale to other psychological measures such as BJSQ. However, we can discuss its appropriateness from a theoretical perspective. Conventional research regarding work and workplace improvements has been oriented towards acquiring findings that could be applied broadly to workers irrespective of their context [24]. This orientation justified the researchers’ determination of which items should be included for measurement and gave the participants (workers) a passive role. However, individual workers are beings whose ‘processes are psychologically channelised by the ways [they] anticipate events’ and analyse the similarities and differences in events to find meaning, a theoretical perspective known as organised the theory of personal constructs [25]. The SWING instrument adheres to this theory by using respondent-generated measurement items whose meanings the respondents rank in terms of sufficiency level and weight balance. We are convinced that SWING, which elicits workers’ subjective workplace-improvement needs, can be supported by its appropriateness from the perspective of this theory.

SWING was inspired by O’Boyle et al.’s Schedule for the Evaluation of Individual Quality of Life-Direct Weighting (SEIQoL-DW), a methodology for measuring the subjective quality of life (QoL) [26]. Although both the SEIQoL-DW and SWING measure sufficiency and weight separately and can calculate index scores, the SEIQoL-DW collects data using interviews; SWING collects data using self-administered questionnaires with simple instructions. The SEIQoL-DW is intended to ascertain the QoL of individuals with disease or disabilities who may have difficulty in answering questions for use in rehabilitation and care [27,28,29]. SWING targets all workers able to answer questions about their subjective workplace-improvement recommendations. SWING differs from SEIQoL-DW in its data collection methods, intended subjects, and post-measurement applications.

### 1.4. Purpose and Hypotheses of the Present Study

SWING is a new method that requires verification. One area where a comprehensive instrument for identifying workers’ decent work needs to be applied is nursing homes in Japan, which have a high turnover rate, raising concerns about the declining quality of care [30]. Therefore, nursing home workers in Japan were considered appropriate as a worker population for the verification of SWING.

This study aimed to verify a new measurement tool, SWING, to elicit workers’ recommendations for practical solutions to workplace problems in nursing homes for the elderly in Japan. This paper describes SWING in detail from the following two perspectives: first, its distribution form, i.e., its relationship with existing measures for workplace improvement, and group and individual SWING applications; second, SWING’s versatility and limitations. For the first perspective mentioned above, we posit the following three hypotheses: (1) the SWING index score has a standard normal distribution form similar to that of SEIQOL-DW [19]; (2) given that the items inductively generated and assessed in SWING may partially be in common with those of previous studies, there is a weak association between SWING and existing measures for workplace improvement; and (3) there is sufficient variation in the results of SWING in terms of item, sufficiency, and weight to determine the priority of workplace-improvement needs.

## 2. Materials and Methods

### 2.1. Setting and Recruitment

We distributed self-administered, non-anonymised questionnaires in October and November 2020 to 70 workers in a nursing home for elderly in Fukushima, Japan. Studies have shown that nursing home workers in Japan are dissatisfied with their workplaces, and the facilities have a high turnover rate [30]. We selected nursing home workers as our target population because we expected they would have (and be willing to discuss) specific recommendations for workplace improvements.

Of the 70 workers, 53 responded to the questionnaire (response rate 75.7%; 53/70). None of the returned questionnaires had any missing data (effective response rate: 75.7%; 53/70). We collected the respondents’ names so we could provide feedback after the survey. However, collecting the respondents’ names is unnecessary if they do not desire feedback.

### 2.2. Measurement

The participants completed a self-administered questionnaire comprising basic demographic questions, the SWING items, and a job-stress scale. The SWING items measured the respondents’ five cues and their corresponding sufficiency level and relative importance. To avoid bias during the analyses, the second author contacted any respondents whose questionnaires were incomplete and helped them rectify the omissions, and the first author conducted the analyses. For the job-stress scale, we used the BJSQ [7], which consists of 80 items measuring workload, physical burden, interpersonal stress, workplace stress, degree of control, skill utilisation, job fitness, and sense of reward. The BJSQ is one of Japan’s most widely used scales for workers’ adaptations and workplace improvements, making it appropriate for comparisons to SWING. The higher the BJSQ value, the more favourable or healthier the condition. The range was 1–4.

### 2.3. Qualitative/Statistical Analysis

We examined the respondents’ basic attributes, SWING index scores, and job stress scores using descriptive statistics. We used the Shapiro–Wilk test to verify whether the SWING index score was distributed normally.

We tested the association of the SWING index score with the basic attributes and total job stress score using bivariate analysis: a *t*-test for sex, an analysis of variance (ANOVA) with multiple comparisons by Tukey’s honestly significant difference (HSD) test for age group, and Spearman’s rank correlation coefficient for job stress score.

The five cues were vital because they represented the core of the workers’ subjective workplace-improvement needs. However, the content of the cues varied widely. Therefore, we used an inductive coding procedure with the open coding method [31], assigning labels to the cues to abstract the underlying meaning, then merging them into larger concepts. The three authors conducted the inductive coding jointly. The first author specialised in qualitative psychology research; the second worked at the nursing home for the elderly (the study’s setting); and the third was an occupational physician. Together, we qualitatively analysed and summarised the sentences of 265 unique cues into 21 workplace-improvement needs categories. We calculated the average sufficiency level and weight for each of the 21 items for the group and individual analyses.

During the analysis for whole subjects, we created a bubble plot illustrating the 21 overall participant recommendations for workplace improvements. We plotted the mean sufficiency level on the vertical axis and the weight on the horizontal axis, forming four quadrants centred on the total mean sufficiency level and weight. The number of respondents who mentioned an item determined the size of the 21 bubbles, and their position indicated their relative importance. The figure graphically represents the workers’ priorities for workplace improvement. The top left category represents important but unmet needs and comprises the following four items: interaction with customers, physical and psychological harassment, rewards and challenging work, and sharing goals and objectives.

We used the participants’ input and index scores, in addition to sufficiency level and weight, to collate the recommendations for the workplace management to implement to ensure a decent work environment. We extracted three cases of participants responses to exemplify how to utilise individual results of SWING for workplace managers.

We performed all the statistical analyses in December 2021 using IBM SPSS Statistics for Windows, Version 28.0 (Armonk, New York, NY, USA: IBM Corp.), setting the significance level at 0.05 (5%).

### 2.4. Ethics

The study protocol was approved by the Ethics Committee of the Fukushima Medical University, Fukushima, Japan (Application No. 2020-123).

## 3. Results

Table 2 shows that most of the respondents were aged in their 40s. The mean SWING index score was 51.5, and the median job stress score was 2.275. As shown in Appendix A, the normality of the SWING index score distribution was confirmed (*p* = 0.145).

As shown in Table 3, the SWING index score was significantly associated with age (*p* = 0.001) and job stress (*p* = 0.029); the correlation coefficient between the SWING index score and job stress was 0.3.

Table 4 details the descriptive textual information, summarised as 21 workplace-improvement needs items based on 265 cues. The items’ content was diverse. For example, there were multiple items related to occupational health management (e.g., workload balance, work flexibility, number of staff, and work environment) and items emphasising the psychosocial aspects of the work environment (e.g., rewarding and challenging work).

Table 5 describes information such as the degree of sufficiency, weight balance, and number of people who mentioned each need item. The bubble plot using said information is shown in Figure 1, and the overlapped items placed nearby the intersection of axes are depicted in detail in Figure 2.

As shown in Table 6, Case 1 showed large differences in both the sufficiency level and weight values, Case 2 had gaps in the sufficiency level, and Case 3 had gaps in the weight values. The SWING index scores for Cases 1–3 were 26, 50, and 73, respectively.

## 4. Discussion

### 4.1. SWING Results and Practical Implications

This study developed and verified a new measurement method, SWING, that enables workers to self-generate recommendations for improving their workplace and rank them according to their current provision level (sufficiency) and relative importance (weight). Consistent with our hypotheses, SWING had a standard normal distribution in its index score, a weak association with an existing scale, BJSQ, and variation in measurement results. From these results, we assert that the SWING instrument effectively extracted unique workplace-improvement needs and precisely revealed the degree of sufficiency and importance through item weighting. SWING has an advantage over other workplace-oriented psychological scales such as the BJSQ because SWING’s self-generated items allow for content-based measurement that can fit any occupation or workplace, making it highly versatile.

Most of the participants in this study were aged in their 40s, which typifies Japan’s working population [32]. Both the SWING index score and job stress score were about half of the maximum values. These moderate ratings suggest that the participants did not identify extreme deficiencies in workplace environments or high levels of job stress. The normal distribution of the SWING index scores will provide many options in further studies for statistical analysis.

The SWING index scores decreased as the respondents’ age increased. This may be because the older workers’ (60+) greater cumulative experience made them more aware that there was room for improvement in their work or workplaces, and less tolerant of conditions they did not associate with ‘decent work’ [1]. Although the SWING index scores were explicitly associated with job stress, the correlation coefficient was low (0.3). As shown in Table 4, SWING extracted a wide range of workplace-improvement needs, but not all of those needs were specifically associated with job stress. Thus, SWING can identify a broader spectrum of concerns than instruments with a purely psychological focus, like the BJSQ.

We identified 21 categories of workplace-improvement needs. Although achieving the ILO’s ideal ‘decent work’ [1] requires addressing all the concerns, a key benefit of the SWING tool is that it automatically prioritises the workers’ concerns. This may help organisations develop implementation plans that allocate resources (e.g., money, personnel, time) to optimise workplace improvements. For example, the item ‘commuting conditions’ may reflect policies or infrastructure issues outside the purview of the specific workplace; thus, the organisation might make advocating for changes in those areas a lower priority than changes they could personally implement, such as arranging carpools or installing parking areas. In contrast, the items ‘workload balance’, ‘work flexibility’, ‘number of staff’, and ‘work environment’ would be workplace-specific and critical; therefore, the organisation would consider them a high priority. In healthcare organisations like the setting of this study, this may suggest the need for programs to implement risk-based activities and occupational health activities (e.g., the Occupational Health Management System or the Five Management system’s Role for Occupational Physicians) [33]. The priorities identified by the SWING analysis may help the organisations allocate their resources to optimise workplace improvements.

Bubble plots are a good tool for visualising the workplace-improvement needs of a large target population, such as an entire workplace. Figure 1’s quadrants visually indicate each item’s importance and current sufficiency level, readily identifying the most pressing concerns: the items respondents consider most important and the most lacking appear in the upper left quadrant. For example, the item ‘interaction with customers’ in the upper left quadrant emphasises its importance; previous studies have reported that such interactions in nursing homes were associated with increased burnout and job stress [34,35].

Table 4 suggests that patients’ pleasant feedback (e.g., ‘smiles’ in the item ‘interaction with customers’) can encourage and motivate workers. Thus, organisations such as this nursing home may need to create an atmosphere that improves client–staff interactions by setting shared expectations, maximising both parties’ satisfaction, and establishing best practices for routine interactions (e.g., respecting patients’ preferences for food or personal space). One previous study found that humour-enhancement training for both residents and staff in elderly care facilities increased communication and care quality [36].

One area of concern was ‘physical or psychological harassment’. Although it was beyond the scope of this study to quantify or qualify harassment in the workplace, the ILO’s Decent Work report [1] explicitly states that workplace harassment is unacceptable. Harassment is a human rights issue because it decreases physical and mental health and reduces the quality of management [37,38]. Occupational health professionals, organisations, and mental health support professionals should collaborate to jointly establish effective anti-harassment policies with equitably applied sanctions for violations [39].

Another item was ‘rewarding and challenging work’. Previous studies have reported that the absence of work rewards (financial, emotional, or symbolic) was associated with turnover intentions [40]; challenging work lacking support from colleagues was associated with low job satisfaction [41]; and appropriate rewarding or challenging job demands increased work engagement and mental health [42]. Thus, organisations may lower turnover by rewarding their workers and providing emotionally satisfying and challenging jobs. However, the best timing for changes focused on enhancing work engagement depends on the workers’ career trajectories; one study found that workers who were just starting their careers desired increasingly challenging jobs, whereas mid-career workers were more resistant to major changes to their jobs and responsibilities [43]. Thus, organisations need to ensure that any changes they make to improve workers’ long-term job satisfaction and engagement are sensitive to the workers’ short-term needs, which may mean rewarding workers for accepting greater responsibilities or adopting new procedures or protocols and implementing substantial changes incrementally.

The item ‘sharing goals and objectives’ emphasised the need for organisations to share their goals and objectives. Working in the absence of shared goals and objectives can lead to role ambiguity, creating workplace dysfunction [44] or high turnover due to emotional exhaustion [45]. Employers and workers regularly communicate their overall goals and objectives. Research recommends that organisations should provide complete and fair information through employee-centred communication systems and encourage employees to participate in determining the organisation’s goals and objectives [46]. Such communication strategies may ensure decent workplace environments.

We considered solutions regarding the three cases extracted for exemplification with a deep understanding of individual worker’s workplace-improvement needs (Table 6). Although Case 1 showed significant variation in the sufficiency level and weight (importance), the item ‘interaction with customers’ had a markedly low sufficiency level and a high importance level, suggesting the need to prioritise improvements in that area. For example, depending on the Case 1 respondent’s competencies, the nursing home manager may move that person to a role with more or less direct interaction with patients. In Case 2, the sufficiency level varied widely but the weight balance (importance) was evenly distributed. This suggested that the highest priorities for allocating resources for workplace improvements should be in the areas with the lowest sufficiency levels. For example, the items ‘trusts relationships in the workplace’ and ‘rewarding and challenging work’ showed equal sufficiency levels and weight balance. However, creating jobs that were rewarding and challenging would be a relatively easy and short-term improvement, whereas trust building takes time and ceaseless effort. In Case 3, the weights varied, so the needs fulfilment for the items with the highest weighting (importance)—‘desire for recognition’ and ‘work environment’—should be the top priorities. However, it is important to view all the cues before allocating resources. Case 3 had a higher index score than Cases 1 and 2, so those items should take priority.

### 4.2. SWING’s Versatility

Although the subjects of this study were middle-aged and elderly workers in a nursing home, SWING can be applied to any worker in any organisation, providing they can understand the items and respond. Although this study used a paper-based questionnaire, SWING can also be administered through an electronic or online survey as long as simple completion instructions are included.

The key points of using SWING in practice are summarised as follows. First, the procedure for summarising the cues into workplace-improvement needs may seem challenging because it involves qualitative analysis. We used the open coding method; however, any analysis method can be employed if it properly summarises qualitative data. Although our analysts had experience working with qualitative research, most organisation’s’ human resources professionals and managers can do this. In the case of our study setting, a nursing home in Japan, any of the occupational health professionals (e.g., occupational health nurses, health managers) can learn quickly from a SWING instruction manual how to conduct these analyses. The analysts should avoid excessive abstraction and summarise the cues while retaining nuances as close to the raw data as possible. SWING has no fixed number of cues or categories. Our study of a single nursing home in Japan had 265 cues divided into 21 types; other organisations may require more or fewer cues. Because of the inductive nature of qualitative analyses, the final number of needs items cannot be determined in advance. As a practical matter, too many need items cannot be usefully displayed in a bubble plot, as the excessive overlays would reduce its utility. We recommend a target maximum of 25 types.

Second, SWING can be used both as a factor and outcome in occupational health studies. For example, as a factor, SWING may use turnover as an outcome and examine its association and causality. This example has a design that explains or predicts the occurrence of turnover by the extent to which work-improvement needs are fulfilled, using needs items as necessary. As an example of using SWING as an outcome, researchers can explore the association–causality of work-improvement interventions, such as implementing sophisticated occupational health management. In this case, SWING would measure the degree of satisfaction with the workplace before and after the improvement interventions. Just as the original SEIQoL instrument is often used to examine the effects of care or rehabilitation [27,28,29], SWING can be used in longitudinal and cross-sectional studies to measure changes in index scores and item variability after interventions.

Third, SWING can be used as a worker-reported outcome (WRO) that applies the concept of patient-reported outcome (PRO) to workers. PROs have the merit of facilitating discussion between clinicians and patients, thus contributing to improved disease management [47], cost-effectiveness analysis, and health policy development [48]. Similarly, SWING as a WRO may promote employer–worker dialogue and contribute to workplace improvement, and provide information on occupational health policy.

### 4.3. Limitations

This study has two limitations. First, the performance of SWING was verified in a nursing home for the elderly in Japan. Although SWING can be easily applied to any healthcare setting or an organisation (e.g., manufacturing, education, marketing, retail, etc.) in principle, further verification should be conducted in different workplaces to prove its effectiveness. Such further verification may contribute to the modification of SWING in accordance with the organisation where it is used. Second, the present study employed a cross-sectional research design; thus, no verification of SWING was conducted in a longitudinal study. SWING may be applicable to both cross-sectional and longitudinal studies as it can be used as a factor or outcome in the research design.

## 5. Conclusions

This study developed and verified a new measurement method, SWING, which enables workers to self-generate recommendations for improving their workplace and rank them according to their current provision level (sufficiency) and relative importance (weight). SWING extracted items for workplace-improvement needs unique to a specific organisation and its workers, clarifying groups’ and individuals’ perceptions of the sufficiency level and weight balance of the recommendations using tables and graphs (e.g., bubble plots), and providing an index score summarising the overall needs. Because the participants self-generate the areas needing improvement and rank them according to how well they are being addressed and their relative importance in a decent work environment, SWING can be applied to any occupation or workplace when attempting to determine the priority of workplace-improvement needs. From a practical standpoint, a maximum of 25 items to measure types of workplace-improvement needs is recommended, and SWING can be used both as a factor and an outcome measure. SWING as a worker-reported outcome may facilitate employer–worker dialogue, elicit better workplace-improvement needs, and provide fundamental knowledge for occupational health policy.

## Figures and Tables

**Figure 1 ijerph-19-01671-f001:**
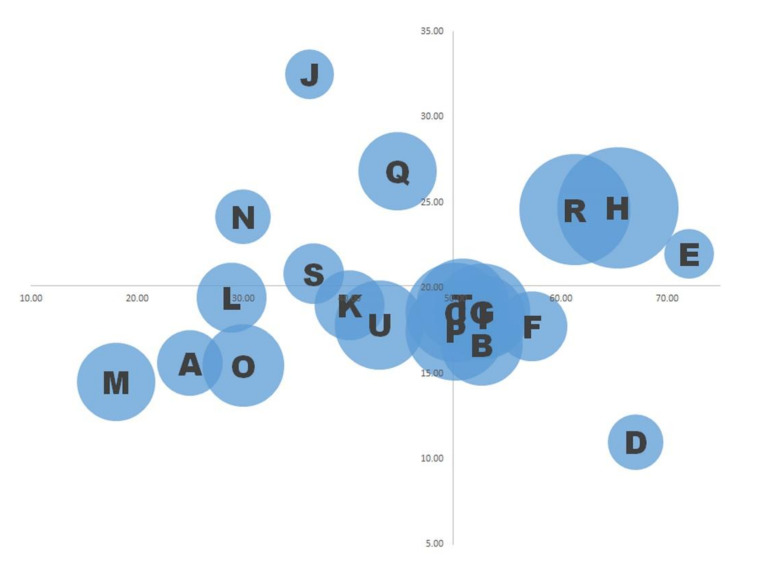
Bubble plot of work-improvement needs items in SWING for whole subjects. The vertical axis is mean weight balance and horizontal axis is mean sufficiency level. Centred on the intersections set by the overall mean weight and sufficiency, four patterns, such as the upper left quadrant with high importance and low sufficiency, were obtained. The items that were mentioned by many people are depicted as large, whereas items that were not mentioned much are depicted as relatively small bubbles. The items placed around the intersections and overlapped are B, C, G, I, P, and T. Those items were not well characterised in terms of both sufficiency and weight in this study. Due to the limit of the available space, the items are depicted by following symbols. A: Balance of workload; B: Benefit package; C: Communication with colleagues; D: Commuting conditions; E: Desire for recognition; F: Flexibility of work; G: Holidays; H: Human relations; I: Information sharing; J: Interaction with customers; K: Legal compliance and rules at work; L: Number of staff; M: Personnel evaluation; *n*: Physical/psychological harassment; O: Possibility of self-growth; P: Relationship of trust in the workplace; Q: Rewarding and challenging work; R: Salary; S: Sharing of goals and objectives; T: Trusted person to confide in; U: Work environment.

**Figure 2 ijerph-19-01671-f002:**
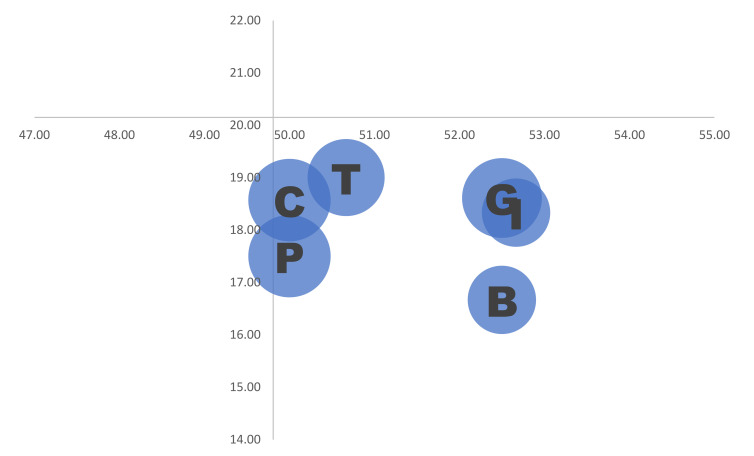
An enlarged illustration near the intersection of the axes of Figure 1. The overlapped items in Figure 1, namely B, C, G, I, P, and T, are visualised.

**Table 1 ijerph-19-01671-t001:** Examples of psychological scales/instruments to enhance decent work.

Name	Purpose	Advantage/Disadvantage	Measurement
Minnesota Job Satisfaction Questionnaire [2]	To provide more specific information about particular aspects of the job that individuals find rewarding rather than measuring general job satisfaction.	This questionnaire can examine the status of job satisfaction from both the individual (intrinsic) and environmental (extrinsic) perspectives. However, given the diversity of workplaces and occupations, a myriad of additional studies is needed to ensure the generality of the items in this scale.	Twenty aspects of job satisfaction, e.g., ability utilisation, achievement, and activity, are measured using a five-point Likert scale.
Job Satisfaction Survey [3]	To measure the job satisfaction of workers, especially those in welfare, public, and non-profit sector organisations.	Based on the theory that job satisfaction is an emotional or attitudinal response to work, this scale is composed of items that can be applied to any occupation. Contrarily, such items do not fully reflect the improvement needs in individual workplaces.	Job satisfaction composed by the nine factors, e.g., pay, promotion, and supervision, are measured using a six-point Likert scale.
Multidimensional Scale of Perceived Social Support [4]	To provide a brief rating to identify the social support that an individual receives from family, friends, and significant others.	Although this scale can evaluate the level of social support received from others such as friends and family, the relationships with others such as pets or psychotherapists are not examined because such relationships are outside the scope of this scale.	Perceived social support from family, friends, and significant others is measured using a seven-point Likert scale for 12 items.
Maslach Burnout Inventory-General Survey [5]	To measure the status of burnout as workers’ response to job-related stress.	This scale takes the theoretical position that burnout is a crisis that occurs in the relationship between self and work, instead of as a result of interpersonal relationships, and thus can be applied to workers in a variety of industries. However, the appropriateness of the factor structure of this scale is debatable [14].	Frequencies to the burnout-related questionnaire items consisting of exhaustion, cynicism, and professional efficacy sub-factors are measured using a seven-point Likert scale.
Job Content Questionnaire [6]	To measure important workplace problems that are often overlooked because they are difficult to assess in terms of their job content.	This questionnaire is based on a theory that explains occupational stress in terms of an imbalance between job responsibilities and discretionary authority, and the items are comprehensive. Contrarily, when this scale is used for a new population, it is necessary to verify whether such items can measure job content appropriately.	This scale contains of 49 items in five scales: decision latitude, psychological demands and mental workload, social support, physical demands, and job insecurity, and measures subjects’ work environment using four-point Likert scale.
Brief Job Stress Questionnaire [7,8,9]	To achieve work improvement for both individuals and workplaces through the prevention of mental health problems among workers.	Although the questionnaire includes items on all aspects of physical, mental, and social health, it does not measure the subjective importance/weight of each item or factor.	Questions regarding job stressors, psychological stress reactions, and social supports are asked using a four-point Likert scale.
Copenhagen Psychosocial Questionnaire [10]	To improve and facilitate research and practical interventions at workplaces by assessing psychosocial factors at work, stress, employees’ well-being, and some personality factors.	This scale can examine both internal factors, e.g., personality, and environmental factors, e.g., time for tasks, by measuring psychosocial factors related to work in three dimensions: workplace, work-individual interface, and individual. However, the subjective importance of each question item is not included in the scope of this scale.	This scale measures an individual’s internal and environmental factors. Although the method of measurement varies depending on the question, most of the responses are measured using a five-point Likert scale.
Mental Health Improvement & Reinforcement Research of Recognition [11]	To obtain information from objective assessments of mental health among workers to understand the state of the workplace and to make improvements.	Because this questionnaire consists of questions about working conditions and environment, it is easy for researchers to gain an objective understanding of workplace conditions, thus leading to improvement. Contrarily, subjective assessment of the importance or relevance of questionnaire items assigned by individual workers on is not possible.	Using a four-point Likert scale, the degree of need for improvement in items related to mental health in a workplace, e.g., workload and supervisor behaviour, is evaluated.
Perceptions of Fair Interpersonal Treatment scale [12]	To assess employees’ perceptions of interpersonal treatment in their workplace, that is, their sense of fairness.	This scale can extract harassment and oppression in interpersonal relationships. Contrarily, this scale assumes a two-party relationship between the worker and the supervisor or co-worker; thus, it is not able to examine the content and quality of the work itself.	This scale is composed of supervisor and co-worker factors, and the degree of applicability to the items is asked and scored using a three-point Likert scale.
Quality of Worklife Questionnaire [13]	To examine the quality of work life by examining a wide range of organisational issues.	This questionnaire examines a wide range of factors related to worker safety and health such as job level, working hours, and culture. However, the extent to which workplace-improvement needs are subjectively met is not explored in this questionnaire.	For each of the nine aspects of work-life, e.g., job level, culture/climate, and health outcomes, subjects choose the ones that apply and rate them on a Likert scale or describe them.

Note: the scales/instruments were ordered by mentions in the text.

**Table 2 ijerph-19-01671-t002:** Characteristics (*n* = 53).

Variables	Values
Gender	
Male	18 (34)
Female	35 (66)
Age (mean ± SD)	46 ± 13.5
<30	6 (11.3)
30–39	12 (22.6)
40–49	18 (34.0)
50–59	7 (13.2)
≥60	10 (18.9)
SWING Index Score (mean ± SD)	51.5 ± 21.2
Job stress; median (25–75 percentile)	2.275 (2.21–2.42)

Note: gender and age group were described by *n* (%).

**Table 3 ijerph-19-01671-t003:** Associations of SWING index score with variables.

Variables	SWING Index Score	*p*-Value
Gender		0.370
Male	47.8	
Female	53.4	
Age group		0.001 *
<30 ^a^	73.9	
30–39 ^b^	53.7	
40–49 ^c^	54	
50–59 ^d^	50.6	
≥60 ^e^	31.4	
Job stress	0.3	0.029 *

Note: Statistical significance was examined using t-test for gender, ANOVA for age group, and Spearman’s rank correlation coefficient (ρ), and indicated by *. Using the symbols “a” to “e” for categories of age group, the multiple comparison of ANOVA for age group indicated the significant differences: a = c > e.

**Table 4 ijerph-19-01671-t004:** Work-improvement needs items generated by summarising the cues regarding pleasant work/workplace.

Work-Improvement Needs Items	Example
Balance of workload	Appropriate work volume and workload; The number of customer appropriate to the situation at the workplace.
Benefit package	Welfare; Good support for childcare; Enjoyment beside work such as company trip.
Communication with colleagues	Good communication in the workplace; Reflection of opinions from each worker regardless of workplace hierarchy; No backbiting, swearing, or whispering.
Commuting conditions	Commuting distance and hours; Presence of convenience store nearby; Location of workplace; Distance from home to workplace.
Desire for recognition	Mutual respect among workers; Feeling that I am needed by the company.
Flexibility of work	Security of private time; Work-life balance; No unreasonable working hours; Less overtime work; Controllability of work and rest;
Holidays	Ease of taking vacations; Ease of taking paid holidays; Many holidays in year end and new-year, and summer/winter vacations.
Human relations	Good human relations; Harmony; Cheerful atmosphere; Stress-free relations in workplace; Polite manner such as greetings.
Information sharing	Ease of opinion exchange; Each staff member understands his or her own position and role, and performs his or her work; Sharing new information, knowledge, and methods for work.
Interaction with customers	Positive feedback from customers; Smiling faces of customers motivate me.
Legal compliance and rules at work	Legally appropriate working hours; Uniformed and clear work flow; Strict adherence to food hygiene.
Number of staff	Sufficient number of staff; Appropriate Employment management.
Personnel evaluation	Fairness; Transparency of the evaluation system; Presence of evaluation criteria based on effort, innovation, productivity in individuals, instead of attendance number of work.
Physical/psychological harassment	An environment free of moral harassment; No bullying; Equal treatment without pressure or imposition.
Possibility of self-growth	Good education program; Environment where skills and knowledge can be improved; Well-developed human resource development system.
Relationship of trust in the workplace	Trust among staff members; No sense of distrust; No selfish attitude; Helping each other; Working together in case of trouble.
Rewarding and challenging work	Job that I like, am good at, and want to do; Job satisfaction; Motivating and rewarding; Rewarding or a sense of accomplishment in work.
Salary	Satisfactory salary and wage levels; Properly paid salary; Wages commensurate with the nature of the work; Compensation for the overtime hours.
Sharing of goals and objectives	Shared values in the company; Agreeable management policy of the facility; Common goals; Leader who can put him/herself in our position.
Trusted person to confide in	Ease of consultation about both work and non-work matters to colleague and/or supervisor; Respectable and reliable supervisor; Accurate advice that corrects mistakes and leads to success.
Work environment	Cleanliness of workplace; Well-equipped facilities; Maintenance; Air conditioning; Necessary supplies well-stocked.

**Table 5 ijerph-19-01671-t005:** Work-improvement needs items, sufficiency level, weight balance, and total number of individuals who mentioned the corresponding item in SWING.

Item	Sufficiency Level	Weight Balance	Total Number of Individuals Who Mentioned (*n* = 53)
Overall mean ± SD	49.8 ± 26.4	20.2 ± 10.7	
Balance of workload	25.00	15.63	7 (13.2)
Benefit package	52.50	16.67	11 (20.8)
Communication with colleagues	50.00	18.57	16 (30.2)
Commuting conditions	67.00	11.00	5 (9.4)
Desire for recognition	72.00	22.00	4 (7.5)
Flexibility of work	57.22	17.78	8 (15.1)
Holidays	52.50	18.61	15 (28.3)
Human relations	65.29	24.71	24 (45.3)
Information sharing	52.67	18.33	11 (20.8)
Interaction with customers	36.25	32.50	4 (7.5)
Legal compliance and rules at work	40.00	19.00	8 (15.1)
Number of staff	28.89	19.44	8 (15.1)
Personnel evaluation	18.00	14.50	10 (18.9)
Physical/psychological harassment	30.00	24.17	5 (9.4)
Possibility of self-growth	30.00	15.45	11 (20.8)
Relationship of trust in the workplace	50.00	17.50	16 (30.2)
Rewarding and challenging work	44.55	26.82	10 (18.9)
Salary	61.25	24.58	20 (37.7)
Sharing of goals and objectives	36.67	20.83	6 (11.3)
Trusted person to confide in	50.67	19.00	14 (26.4)
Work environment	42.86	17.86	13 (24.5)

Note: Sufficiency level and weight balance were described by mean, and total number of individuals who mentioned was described by *n* (%).

**Table 6 ijerph-19-01671-t006:** Examples of results in SWING for individual analysis.

Cases and Items	Sufficiency Level	Weight Balance	Score (Index Score)
Case 1			
Benefit package	60	20	12
Human relations	50	10	5
Interaction with customers	0	40	0
Possibility of self-growth	30	10	3
Work environment	30	20	6
Index Score			26
Case 2			
Flexibility of work	70	20	14
Personnel evaluation	90	20	18
Possibility of self-growth	50	20	10
Relationship of trust in the workplace	20	20	4
Rewarding and challenging work	20	20	4
Index Score			50
Case 3			
Communication with colleagues	90	10	9
Commuting conditions	95	10	9.5
Desire for recognition	90	30	9.5
Holidays	95	10	27
Work environment	90	20	18
Index Score			73

Note: The items were placed in alphabetical order regardless of their appearance order.

## Data Availability

The data presented in this study are available on request from the corresponding author. The data are not publicly available because the data include personally identifiable information.

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
