# Peer review of "The Systematic Workplace-Improvement Needs Generation (SWING): Verifying a Worker-Centred Tool for Identifying Necessary Workplace Improvements in a Nursing Home in Japan"

_ijerph, 2022, doi:10.3390/ijerph19031671_

Round 1
Reviewer 1 Report
Article Developing a worker-centered tool for identifying necessary
workplace improvements: Systematic Workplace-Improvement Needs Generation (SWING) is interesting but cannot be published in this form because:
1. The introduction is brief
2. The hypotheses of the study are not formulated
3. The sample is extremely small
4. Conclusions are not based on the results of the study
5. No study limitations
6. The applied questionnaire is attached in Japanese
Author Response
Dear Reviewers and Editor,
We wish to express our appreciation to you for your insightful comments, which we believe have helped us to improve our manuscript.
The revised text was highlighted in yellow. Although the Journal’s instruction recommended the use of “Track Changes” function on Word, it didn't work properly probably due to the amount of texts. Please allow for this highlighting.
For footnotes, the URL of Supplementary Materials have been assigned and thus we added.
[Responses for Reviewer 1]
- The introduction is brief
Response: The reviewer’s comment is correct. To clarify the novelty of this study compared to previous studies, we have added a detailed review of the literature in the Introduction section, using Table 1 (p.2-3). In this review, we exemplified 10 psychological scales that are considered to be related to decent work and summarized their characteristics, advantages and disadvantages. We also have added the following text to indicate the methodological problems of previous studies (p.1 and 4): “These previous studies have in common the methodological fact that the subjects answered questionnaire items that the researchers had judged to be important and were thus employed in their respective scales. …However, these procedures and measurements may have the following methodological limitations.”
- The hypotheses of the study are not formulated
Response: The reviewer’s comment is correct. The study hypotheses have been described in “Purpose and hypotheses of the present study” subsection in Introduction, using the following sentences (p.5, lines 155-161): “we posit the following three hypotheses: 1) the SWING index score has a standard normal distribution form similar to that of SEIQOL-DW[19]; 2) given that the items inductively generated and assessed in SWING may partially be in common with those of previous studies, there is a weak association between SWING and existing measures for workplace improvement; and 3) there is sufficient variation in the results of SWING in terms of item, sufficiency, and weight to determine the priority of workplace improvement needs.”
The relationships between study results and hypotheses have been mentioned in Discussion section (p.10, lines 277-279): “Consistent with our hypotheses, SWING had a standard normal distribution in its index score, a weak association with an existing scale, BJSQ, and variation in measurement results.”
- The sample is extremely small
Response: Thank you for your suggestion. The adequacy of sample size is one of the core questions to our SWING, and thus the detailed explanations are required. The subsection “SWING’s specifications” was moved from Materials and Methods section to Introduction, and we have added the following texts in second paragraph of that subsection (p.4, lines 82-96): “…Accordingly, the researcher needs to group these descriptions inductively/qualitatively. Because of the nature of qualitative analysis, the SWING method will yield different workplace improvement needs items de-pending on the workplace where the survey was conducted, or more precisely, the workers who cooperated in the survey. Given that this trait, which results from 'cues' and consequent 'items' in the SWING method that vary depending on the workplace, consistency or reproductivity of such 'cues' and 'items' is not supported by a (large) sample size of subjects, nor by a criterion of reliability such as test-retest reliability, where the quality of the measurement is ensured by the similarity of the results between surveys. Instead, the appropriateness of SWING as a method should be explored in terms of its data collection procedures: an inductive generation of items that would contribute to the practice of workplace improvement. SWING can be used by any organisation regardless of the number of employees and provides ideas for workplace improvements that are considered to be best for individual workplaces.”
Note that the scientific nature of SWING lies not in the repetition of the same results across large or different samples, but in the procedure of inductively/qualitatively generating and evaluating the categories themselves. The purpose of SWING is to generate workplace improvement needs that are specific to the workplace, instead of extracting robust categories common in workplaces. As explained above, the "results" of SWING can be different for each workplace where SWING is conducted. SWING is instrument to generate workplace improvement needs that are specific to the workplace. Therefore, it is assumed that (small/large) sample size is not an issue from theoretical perspectives in SWING.
- Conclusions are not based on the results of the study
Response: The reviewer’s comment is correct. We have revised the Conclusions section based on the results of the study as follows (p.14, 5. Conclusions section): “This study developed and verified a new measurement method, SWING, which enables workers to self-generate recommendations for improving their workplace and rank them according to their current provision level (sufficiency) and relative importance (weight). SWING extracted items for workplace improvement needs unique to a specific organisation and its workers, clarifying groups’ and individuals’ perceptions of the sufficiency level and weight balance of the recommendations using tables, graphs (e.g. bubble plots) and providing an index score summarising the overall needs. Because the participants self-generate the areas needing improvement and rank them according to how well they are being addressed and their relative importance in a decent work environment, SWING could be applied to any occupation or workplace when attempting to determine the priority of workplace improvement needs.”
- No study limitations
Response: The reviewer’s comment is correct. We have added the Limitations subsection in Discussion section and described the following study limitations (p.14, Limitations subsection): “This study has two limitations. First, the performance of SWING was verified in a nursing home for the elderly in Japan. Although SWING could be easily applied to any healthcare setting or an organisation (e.g. manufacturing, education, marketing, retail, etc.) in principle, further verification should be conducted in different workplaces to prove its effective-ness. Such further verification may contribute to the modification of SWING in accordance with the organisation where it is used. Second, the present study employed a cross-sectional research design; thus, no verification of SWING was conducted in a longitudinal study. SWING may be applicable to both cross-sectional and longitudinal studies as it can be used as a factor or outcome in the research design.”
- The applied questionnaire is attached in Japanese
Response: Thank you for your suggestion. The SWING questionnaires both Japanese and English versions have been uploaded as Supplementary Materials of this study in the following URL: https://doi.org/10.5281/zenodo.5806824
Please find English version of SWING, located with the filename “Supplement2_SWING_ENG.pdf” from the URL above.
The references of English version of the Brief Job Stress Questionnaire which we used in the study have also been indicated as references section with number [8] and [9].
Reviewer 2 Report
Formal comments:
- The title is too long and complicated. It is not usual to explain the abbreviation in title and there is not included information that the article is mainly about verification of this tool in nursing homes in Japan.
- The Figure 1 is in this zoom unclear. Some of the “bubbles” are covered and unreadable.
Content comments:
- The information presented in the article is inaccurate. There are available other tools to evaluate job stress i.e. the Job Stress Assessment Scale, Perceived Social Support Scale, Minnesota Job Satisfaction Questionnaire and Maslach Burnout Inventory-General Survey, or tools for job satisfaction assessment as OMS Assessment questionnaires. It will be useful to extend the Introduction (it could be the separate chapter with Literature review) and include in this part overview of available tools of job stress or satisfaction assessment, their principles, advantages and disadvantages. Based on that it is possible to present the benefit or developed assessment tool.
- There is unclear structure of the article. The chapter “Introduction” covers presentation of the aim of the article and circumstances and very short literature review. In the chapter 2 the methodology of the research is presented, but also the overview of other assessment tools based on which the SWING tool have been developed. These information about the job stress assessment tools will be better located in the Introduction (or exactly in new chapter focused on Literature Review). In the chapter “Materials and Methods” the proposal of the developed assessment tool is described and methodology of the research in nursing homes. It will be clearer if in the first part will be in separate chapter presented the proposal of developed assessment tool in detail and generally (with usage in any type of industry or services) and in the second part description of methodology of verification of this tool in case study of nursing homes in Japan.
Summary:
The proposed tool for identification of necessary workplace improvements, modified based on the specifics of the analysed workplace represents an innovative approach. It does not work with rigid job stress indicators and in accordance to specifics of different workplaces chooses different indicators and change their weight based on view of employees and their importance in different environments.
If it is the first presentation of this tool in literature it is necessary to describe it in detail, that is mean in the way to be able based on that repeat this method in practice. The current description is not satisfied.
The second part of the article brings on one side the verification of this model, and on the other side, the results of the analysis of job satisfaction in nursing homes in Japan. So there is presented the usage of the tool and analysis of job satisfaction in one area. The results from the nursing home are interesting, but for verification will be useful to compare the results from the different workplaces. This is the limit of the article. Based on such comparison there could be visible the declared benefits – the modification of the tool SWING in accordance with area, where it is used.
The recommendation is to modify the structure of the article (to include and extend the literature review), to divide the proposal and description of the tool itself (SWING) and the methodology of the research in nursing homes. It will be useful to describe the SWING in detail. For verification of the tool it will be useful to extend the verification on other area/s (workplaces with different characteristics, i.e. from industry, other type of services, with different type or structure of employees) instead of the long discussion. There is not clear what is the main goal – to present the new assessment tool or to present the result of the research in area of nursing homes in Japan and specify the most important topics impacted the job satisfaction of employees?
Author Response
Dear Reviewers and Editor,
We wish to express our appreciation to you for your insightful comments, which we believe have helped us to improve our manuscript.
The revised text was highlighted in yellow. Although the Journal’s instruction recommended the use of “Track Changes” function on Word, it didn't work properly probably due to the amount of texts. Please allow for this highlighting.
For footnotes, the URL of Supplementary Materials have been assigned and thus we added.
[Responses for Reviewer 2]
Q1. The title is too long and complicated. It is not usual to explain the abbreviation in title and there is not included information that the article is mainly about verification of this tool in nursing homes in Japan.
Response: The reviewer’s comment is correct. We revised the title to “The Systematic Workplace-Improvement Needs Generation (SWING): verifying a worker-centred tool for identifying necessary workplace improvements in a nursing home in Japan” to mention the verification was conducted in a nursing home in Japan and to reduce the redundant expression.
Whereas, we would like to remain the abbreviation “SWING” in the title since such abbreviation may help potential readers find this article in future. The use of abbreviation in a title is often found in the past articles of scales/instruments related to work [A-D], and thus we believe our title is also acceptable.
References for Q1:
[A] Newham JJ, et al. State-trait anxiety inventory (STAI) scores during pregnancy following intervention with complementary therapies. J Affect Disord. 2012;142(1-3):22-30.
[B] Kawada T. Relationship between components of the metabolic syndrome and job strain using a brief job stress questionnaire (BJSQ). Int Arch Occup Environ Health. 2013;86(6):725-726.
[C] Choi YG, et al. A study on the characteristics of Maslach Burnout Inventory-General Survey (MBI-GS) of workers in one electronics company. Ann Occup Environ Med. 2019;31:e29.
[D] Karasek R, et al. The Job Content Questionnaire (JCQ): an instrument for internationally comparative assessments of psychosocial job characteristics. J Occup Health Psychol. 1998;3(4):322-355.
Q2. The Figure 1 is in this zoom unclear. Some of the “bubbles” are covered and unreadable.
Response: The reviewer’s comment is correct. We have added the enlarged illustration as Figure 2 (p.11) to indicate the items placed nearby the intersections of axes. The readability of figure has been improved by this revision.
Q3. The information presented in the article is inaccurate. There are available other tools to evaluate job stress i.e. the Job Stress Assessment Scale, Perceived Social Support Scale, Minnesota Job Satisfaction Questionnaire and Maslach Burnout Inventory-General Survey, or tools for job satisfaction assessment as OMS Assessment questionnaires. It will be useful to extend the Introduction (it could be the separate chapter with Literature review) and include in this part overview of available tools of job stress or satisfaction assessment, their principles, advantages and disadvantages. Based on that it is possible to present the benefit or developed assessment tool.
Response: The reviewer’s comment is correct. To clarify the novelty of this study compared to previous studies, we have added a detailed review of the literature in the Introduction section, using Table 1 (p.2-3). In this review, we exemplified 10 psychological scales related to decent work such as Perceived Social Support Scale, Minnesota Job Satisfaction Questionnaire and Maslach Burnout Inventory-General Survey, and summarized their characteristics, advantages and disadvantages. We also have added the following text to indicate the methodological problems of previous studies (p.1 and 4): “These previous studies have in common the methodological fact that the subjects answered questionnaire items that the researchers had judged to be important and were thus employed in their respective scales. …However, these procedures and measurements may have the following methodological limitations.”
Note that we did not included the Job Stress Assessment Scale by Xing et al in 2012, because the scale was provided only in Chinese and thus not suitable for English readers.
Finally, we have added the following texts to clarify the necessity to develop our new instrument SWING (p.4, lines 61-67): “Consequently, there is no comprehensive instrument for subjectively identifying workers’ needs for workplace improvement. To overcome the methodological limitations mentioned above, that is, inductive generation of items appropriate for individual work-places from the perspective of workers and distinction between sufficiency and weight, it is rational to fundamentally change the measurement procedures. We respond to this demand by proposing a new measurement method: the systematic work-place-improvement needs generation (SWING).”
Q4. There is unclear structure of the article. The chapter “Introduction” covers presentation of the aim of the article and circumstances and very short literature review. In the chapter 2 the methodology of the research is presented, but also the overview of other assessment tools based on which the SWING tool have been developed. These information about the job stress assessment tools will be better located in the Introduction (or exactly in new chapter focused on Literature Review). In the chapter “Materials and Methods” the proposal of the developed assessment tool is described and methodology of the research in nursing homes. It will be clearer if in the first part will be in separate chapter presented the proposal of developed assessment tool in detail and generally (with usage in any type of industry or services) and in the second part description of methodology of verification of this tool in case study of nursing homes in Japan.
Response: The reviewer’s comment is correct. We have moved the subsection “SWING’s specifications” from Materials and Methods section to Introduction (p.4-5), adding sub-subsections such as “Procedures and advantages” and “SWING’s expected contributions to occupational health, theoretical appropriateness, and originality”.
Under the “Procedures and advantages” sub-subsection, for paragraph 1, the existing texts regarding the procedures of SWING have been placed; for paragraph 2, the qualitative/inductive procedures to generate workplace improvement needs and issues of sample size in SWING have been explained in detail by newly added text starting with “For (1) item generation”; for paragraph 3, the advantage of SWING has been summarized by the newly added texts starting with “The distinction between”.
Under the “SWING’s expected contributions to occupational health, theoretical appropriateness, and originality” sub-subsection, the texts which had been placed in Discussion section in previous version of manuscript have been re-located for paragraph 1 and 2. The relationship of SWING and SEIQOL was mentioned in paragraph 3.
The study purpose is development of SWING and its verification in nursing home in Japan, and thus we have added the following underlined texts in subsection “Purpose and hypotheses of the present study” in Introduction section (p.5, lines 145-149): “SWING is a new method that requires verification. One area where a comprehensive instrument for identifying workers’ decent work needs to is nursing homes in Japan, which have a high turnover rate, raising concerns about the declining quality of care [23]. Therefore, nursing home workers in Japan were considered appropriate as a worker population for the verification of SWING.”
We have also revised the purpose of study in addition to hypotheses (p.5, lines 150-161): “This study aimed to verify a new measurement tool, SWING, to elicit workers’ recommendations for practical solutions to workplace problems in nursing homes for the elderly in Japan.”
Q5: The proposed tool for identification of necessary workplace improvements, modified based on the specifics of the analysed workplace represents an innovative approach. It does not work with rigid job stress indicators and in accordance to specifics of different workplaces chooses different indicators and change their weight based on view of employees and their importance in different environments.
If it is the first presentation of this tool in literature it is necessary to describe it in detail, that is mean in the way to be able based on that repeat this method in practice. The current description is not satisfied.
Response: The reviewer’s comment is correct. As we mentioned above, we have added the Table 1 for past literature review to Introduction section for clarifying the differences between the existing instruments and SWING.
Q6: The second part of the article brings on one side the verification of this model, and on the other side, the results of the analysis of job satisfaction in nursing homes in Japan. So there is presented the usage of the tool and analysis of job satisfaction in one area. The results from the nursing home are interesting, but for verification will be useful to compare the results from the different workplaces. This is the limit of the article. Based on such comparison there could be visible the declared benefits – the modification of the tool SWING in accordance with area, where it is used.
Response: Thank you for your suggestion. We have made an independent subsection “Limitations” in Discussion section to show the following limitations including the necessity of further investigation in other workplaces (p.14, lines 420-429): “This study has two limitations. First, the performance of SWING was verified in a nursing home for the elderly in Japan. Although SWING could be easily applied to any healthcare setting or an organisation (e.g. manufacturing, education, marketing, retail, etc.) in principle, further verification should be conducted in different workplaces to prove its effective-ness. Such further verification may contribute to the modification of SWING in accordance with the organisation where it is used. Second, the present study employed a cross-sectional research design; thus, no verification of SWING was conducted in a longitudinal study. SWING may be applicable to both cross-sectional and longitudinal studies as it can be used as a factor or outcome in the research design.”
Q7: The recommendation is to modify the structure of the article (to include and extend the literature review), to divide the proposal and description of the tool itself (SWING) and the methodology of the research in nursing homes. It will be useful to describe the SWING in detail. For verification of the tool it will be useful to extend the verification on other area/s (workplaces with different characteristics, i.e. from industry, other type of services, with different type or structure of employees) instead of the long discussion. There is not clear what is the main goal – to present the new assessment tool or to present the result of the research in area of nursing homes in Japan and specify the most important topics impacted the job satisfaction of employees?
Response: Thank you for your thoughtful suggestion. We have modified the structure to explain (1) the characteristics of SWING in detail and (2) main goal.
For (1) SWING in detail, please confirm the revised texts in SWING’s specifications subsection in Introduction section (p.4-5). For (2) main goal, we clarified the purpose as shown in the following texts (p.5, lines 150-152): “This study aimed to verify a new measurement tool, SWING, to elicit workers’ recommendations for practical solutions to workplace problems in nursing homes for the elderly in Japan. This paper describes SWING in detail from the following two perspectives: first, its distribution form, i.e., its relationship with existing measures for workplace improvement, and group and individual SWING applications; second, SWING's versatility and limitations.”
These purpose and descriptions are consistent with subsections in Discussion section: SWING results and practical implications, SWING’s versatility, and Limitations.
Reviewer 3 Report
Dear Author(s),
Thank you for submitting this manuscript, which addresses a relevant and timely topic. Although the paper reads well, in my opinion, the sample of workers involved in the questionnaire is too small to ensure a generalizability of findings and to give a scientific soundness to your manuscript.
Moreover, the paper lacks a literature section, where to discuss what has already been written on the topic and the reasons why your work is original.
I suggest you to enlarge the sample, include a literature review and then to re-submit the paper.
Best wishes!
Author Response
Dear Reviewers and Editor,
We wish to express our appreciation to you for your insightful comments, which we believe have helped us to improve our manuscript.
The revised text was highlighted in yellow. Although the Journal’s instruction recommended the use of “Track Changes” function on Word, it didn't work properly probably due to the amount of texts. Please allow for this highlighting.
For footnotes, the URL of Supplementary Materials have been assigned and thus we added.
[Responses for Reviewer 3]
- Thank you for submitting this manuscript, which addresses a relevant and timely topic. Although the paper reads well, in my opinion, the sample of workers involved in the questionnaire is too small to ensure a generalizability of findings and to give a scientific soundness to your manuscript.
Response: Thank you for your suggestion. The adequacy of sample size is one of the core questions to our SWING, and thus the detailed explanations are required. The subsection “SWING’s specifications” was moved from Materials and Methods section to Introduction, and we have added the following texts in second paragraph of that subsection (p.4, lines 82-96): “…Accordingly, the researcher needs to group these descriptions inductively/qualitatively. Because of the nature of qualitative analysis, the SWING method will yield different workplace improvement needs items de-pending on the workplace where the survey was conducted, or more precisely, the workers who cooperated in the survey. Given that this trait, which results from 'cues' and consequent 'items' in the SWING method that vary depending on the workplace, consistency or reproductivity of such 'cues' and 'items' is not supported by a (large) sample size of subjects, nor by a criterion of reliability such as test-retest reliability, where the quality of the measurement is ensured by the similarity of the results between surveys. Instead, the appropriateness of SWING as a method should be explored in terms of its data collection procedures: an inductive generation of items that would contribute to the practice of workplace improvement. SWING can be used by any organisation regardless of the number of employees and provides ideas for workplace improvements that are considered to be best for individual workplaces.”
Note that the scientific nature of SWING lies not in the repetition of the same results across large or different samples, but in the procedure of inductively/qualitatively generating and evaluating the categories themselves. The purpose of SWING is to generate workplace improvement needs that are specific to the workplace, instead of extracting robust categories common in workplaces. As explained above, the "results" of SWING can be different for each workplace where SWING is conducted. SWING is instrument to generate workplace improvement needs that are specific to the workplace. Therefore, it is assumed that (small/large) sample size is not an issue from theoretical perspectives in SWING.
- Moreover, the paper lacks a literature section, where to discuss what has already been written on the topic and the reasons why your work is original.
Response: The reviewer’s comment is correct. To clarify the novelty of this study compared to previous studies, we have added a detailed review of the literature in the Introduction section, using Table 1 (p.2-3). In this review, we exemplified 10 psychological scales that are considered to be related to decent work and summarized their characteristics, advantages and disadvantages. We also have added the following text to indicate the methodological problems of previous studies (p.1 and 4): “These previous studies have in common the methodological fact that the subjects answered questionnaire items that the researchers had judged to be important and were thus employed in their respective scales. …However, these procedures and measurements may have the following methodological limitations.”
We have added the following texts to clarify the necessity to develop our new instrument SWING (p.4, lines 61-67): “Consequently, there is no comprehensive instrument for subjectively identifying workers’ needs for workplace improvement. To overcome the methodological limitations mentioned above, that is, inductive generation of items appropriate for individual work-places from the perspective of workers and distinction between sufficiency and weight, it is rational to fundamentally change the measurement procedures. We respond to this demand by proposing a new measurement method: the systematic workplace-improvement needs generation (SWING).”
Round 2
Reviewer 1 Report
The material has been improved in accordance with the suggestions submitted previously.
Author Response
Comment: The material has been improved in accordance with the suggestions submitted previously.
Thank you for your reply. Please note that the texts in Discussion (p.15, lines 464-469) and Conclusions (p.16, lines 492-496) sections have been revised, according to the comment from Reviewer 3 that practical and policy implications and directions for future research should be included. The descriptions in Conclusions section have been supported by the results/mentions of our research, and thus consistent.
Reviewer 2 Report
[Responses for Reviewer 2]
Q1. The title is too long and complicated. It is not usual to explain the abbreviation in title and there is not included information that the article is mainly about verification of this tool in nursing homes in Japan.
Response: The reviewer’s comment is correct. We revised the title to “The Systematic Workplace-Improvement Needs Generation (SWING): verifying a worker-centred tool for identifying necessary workplace improvements in a nursing home in Japan” to mention the verification was conducted in a nursing home in Japan and to reduce the redundant expression.
Whereas, we would like to remain the abbreviation “SWING” in the title since such abbreviation may help potential readers find this article in future. The use of abbreviation in a title is often found in the past articles of scales/instruments related to work [A-D], and thus we believe our title is also acceptable.
Reviewer 2: Modification of the title and explanation of abbreviation presentation in title is acceptable.
References for Q1:
[A] Newham JJ, et al. State-trait anxiety inventory (STAI) scores during pregnancy following intervention with complementary therapies. J Affect Disord. 2012;142(1-3):22-30.
[B] Kawada T. Relationship between components of the metabolic syndrome and job strain using a brief job stress questionnaire (BJSQ). Int Arch Occup Environ Health. 2013;86(6):725-726.
[C] Choi YG, et al. A study on the characteristics of Maslach Burnout Inventory-General Survey (MBI-GS) of workers in one electronics company. Ann Occup Environ Med. 2019;31:e29.
[D] Karasek R, et al. The Job Content Questionnaire (JCQ): an instrument for internationally comparative assessments of psychosocial job characteristics. J Occup Health Psychol. 1998;3(4):322-355.
Q2. The Figure 1 is in this zoom unclear. Some of the “bubbles” are covered and unreadable.
Response: The reviewer’s comment is correct. We have added the enlarged illustration as Figure 2 (p.11) to indicate the items placed nearby the intersections of axes. The readability of figure has been improved by this revision.
Reviewer 2: The including of the Figure 2 with detail of part of Figure 1 solves the problem.
Q3. The information presented in the article is inaccurate. There are available other tools to evaluate job stress i.e. the Job Stress Assessment Scale, Perceived Social Support Scale, Minnesota Job Satisfaction Questionnaire and Maslach Burnout Inventory-General Survey, or tools for job satisfaction assessment as OMS Assessment questionnaires. It will be useful to extend the Introduction (it could be the separate chapter with Literature review) and include in this part overview of available tools of job stress or satisfaction assessment, their principles, advantages and disadvantages. Based on that it is possible to present the benefit or developed assessment tool.
Response: The reviewer’s comment is correct. To clarify the novelty of this study compared to previous studies, we have added a detailed review of the literature in the Introduction section, using Table 1 (p.2-3). In this review, we exemplified 10 psychological scales related to decent work such as Perceived Social Support Scale, Minnesota Job Satisfaction Questionnaire and Maslach Burnout Inventory-General Survey, and summarized their characteristics, advantages and disadvantages. We also have added the following text to indicate the methodological problems of previous studies (p.1 and 4): “These previous studies have in common the methodological fact that the subjects answered questionnaire items that the researchers had judged to be important and were thus employed in their respective scales. …However, these procedures and measurements may have the following methodological limitations.”
Note that we did not included the Job Stress Assessment Scale by Xing et al in 2012, because the scale was provided only in Chinese and thus not suitable for English readers.
Finally, we have added the following texts to clarify the necessity to develop our new instrument SWING (p.4, lines 61-67): “Consequently, there is no comprehensive instrument for subjectively identifying workers’ needs for workplace improvement. To overcome the methodological limitations mentioned above, that is, inductive generation of items appropriate for individual work-places from the perspective of workers and distinction between sufficiency and weight, it is rational to fundamentally change the measurement procedures. We respond to this demand by proposing a new measurement method: the systematic work-place-improvement needs generation (SWING).”
Reviewer 2: Such extension is exactly what I recommended. Based on such overview and explanation of disadvantages of other assessment tools the reader of the article has got sufficient information and could understand the reasons for development of new tool.
Q4. There is unclear structure of the article. The chapter “Introduction” covers presentation of the aim of the article and circumstances and very short literature review. In the chapter 2 the methodology of the research is presented, but also the overview of other assessment tools based on which the SWING tool have been developed. These information about the job stress assessment tools will be better located in the Introduction (or exactly in new chapter focused on Literature Review). In the chapter “Materials and Methods” the proposal of the developed assessment tool is described and methodology of the research in nursing homes. It will be clearer if in the first part will be in separate chapter presented the proposal of developed assessment tool in detail and generally (with usage in any type of industry or services) and in the second part description of methodology of verification of this tool in case study of nursing homes in Japan.
Response: The reviewer’s comment is correct. We have moved the subsection “SWING’s specifications” from Materials and Methods section to Introduction (p.4-5), adding sub-subsections such as “Procedures and advantages” and “SWING’s expected contributions to occupational health, theoretical appropriateness, and originality”.
Under the “Procedures and advantages” sub-subsection, for paragraph 1, the existing texts regarding the procedures of SWING have been placed; for paragraph 2, the qualitative/inductive procedures to generate workplace improvement needs and issues of sample size in SWING have been explained in detail by newly added text starting with “For (1) item generation”; for paragraph 3, the advantage of SWING has been summarized by the newly added texts starting with “The distinction between”.
Under the “SWING’s expected contributions to occupational health, theoretical appropriateness, and originality” sub-subsection, the texts which had been placed in Discussion section in previous version of manuscript have been re-located for paragraph 1 and 2. The relationship of SWING and SEIQOL was mentioned in paragraph 3.
The study purpose is development of SWING and its verification in nursing home in Japan, and thus we have added the following underlined texts in subsection “Purpose and hypotheses of the present study” in Introduction section (p.5, lines 145-149): “SWING is a new method that requires verification. One area where a comprehensive instrument for identifying workers’ decent work needs to is nursing homes in Japan, which have a high turnover rate, raising concerns about the declining quality of care [23]. Therefore, nursing home workers in Japan were considered appropriate as a worker population for the verification of SWING.”
We have also revised the purpose of study in addition to hypotheses (p.5, lines 150-161): “This study aimed to verify a new measurement tool, SWING, to elicit workers’ recommendations for practical solutions to workplace problems in nursing homes for the elderly in Japan.”
Reviewer 2: The structure of the article is now clearer – Introduction presented the theory - the overview of analyzed topic, overview of available assessment tools, reasons for development of tool SWING, the description of this tool and hypotheses, which will be evaluated by verification of developer tool. And in the second part of the article there is presented the verification of the tool in nursing homes in Japan and evaluation of hypotheses and results which the verification brings.
Q5: The proposed tool for identification of necessary workplace improvements, modified based on the specifics of the analysed workplace represents an innovative approach. It does not work with rigid job stress indicators and in accordance to specifics of different workplaces chooses different indicators and change their weight based on view of employees and their importance in different environments.
If it is the first presentation of this tool in literature it is necessary to describe it in detail, that is mean in the way to be able based on that repeat this method in practice. The current description is not satisfied.
Response: The reviewer’s comment is correct. As we mentioned above, we have added the Table 1 for past literature review to Introduction section for clarifying the differences between the existing instruments and SWING.
Reviewer 2: The Table 1 does not bring the solution of this problem, but more detail description in part SWING´s specifications and Procedures and advantages is sufficient, currently is more clear how the utilisation of the SWING based on the different stress indicators in different workplace works.
Q6: The second part of the article brings on one side the verification of this model, and on the other side, the results of the analysis of job satisfaction in nursing homes in Japan. So there is presented the usage of the tool and analysis of job satisfaction in one area. The results from the nursing home are interesting, but for verification will be useful to compare the results from the different workplaces. This is the limit of the article. Based on such comparison there could be visible the declared benefits – the modification of the tool SWING in accordance with area, where it is used.
Response: Thank you for your suggestion. We have made an independent subsection “Limitations” in Discussion section to show the following limitations including the necessity of further investigation in other workplaces (p.14, lines 420-429): “This study has two limitations. First, the performance of SWING was verified in a nursing home for the elderly in Japan. Although SWING could be easily applied to any healthcare setting or an organisation (e.g. manufacturing, education, marketing, retail, etc.) in principle, further verification should be conducted in different workplaces to prove its effective-ness. Such further verification may contribute to the modification of SWING in accordance with the organisation where it is used. Second, the present study employed a cross-sectional research design; thus, no verification of SWING was conducted in a longitudinal study. SWING may be applicable to both cross-sectional and longitudinal studies as it can be used as a factor or outcome in the research design.”
Reviewer 2: It is acceptable to explain the limit of the verification in Discussion and present the plan of further verification in other areas.
Q7: The recommendation is to modify the structure of the article (to include and extend the literature review), to divide the proposal and description of the tool itself (SWING) and the methodology of the research in nursing homes. It will be useful to describe the SWING in detail. For verification of the tool it will be useful to extend the verification on other area/s (workplaces with different characteristics, i.e. from industry, other type of services, with different type or structure of employees) instead of the long discussion. There is not clear what is the main goal – to present the new assessment tool or to present the result of the research in area of nursing homes in Japan and specify the most important topics impacted the job satisfaction of employees?
Response: Thank you for your thoughtful suggestion. We have modified the structure to explain (1) the characteristics of SWING in detail and (2) main goal.
For (1) SWING in detail, please confirm the revised texts in SWING’s specifications subsection in Introduction section (p.4-5). For (2) main goal, we clarified the purpose as shown in the following texts (p.5, lines 150-152): “This study aimed to verify a new measurement tool, SWING, to elicit workers’ recommendations for practical solutions to workplace problems in nursing homes for the elderly in Japan. This paper describes SWING in detail from the following two perspectives: first, its distribution form, i.e., its relationship with existing measures for workplace improvement, and group and individual SWING applications; second, SWING's versatility and limitations.”
These purpose and descriptions are consistent with subsections in Discussion section: SWING results and practical implications, SWING’s versatility, and Limitations.
Reviewer 2: Q7 was the recapitulation of the most important recommendations to authors of the article. After the change of the structure of the article, extension of the Introduction with assessment tools´ overview and SWING´s description, including the Limitations in Discussion and precise explanation of the main goal the article is more understandable and brings an interesting knowhow.
Author Response
Comment #5: The Table 1 does not bring the solution of this problem, but more detail description in part SWING´s specifications and Procedures and advantages is sufficient, currently is more clear how the utilisation of the SWING based on the different stress indicators in different workplace works.
Thank you for your reply. Please note that we have added more detailed description about past literature in Introduction section to explain the background and necessity of developing the SWING, according to the comments from Reviewer 3. This revision may contribute also to the solution of the problem that you reviewer mentioned.
Reviewer 3 Report
Dear Author(s),
Thank you for the revisions and thank you for the explanations regarding the size of the sample and the methodology.
In my opinion, it is still missing a section on the literature review, that can be split from the introduction. Moreover, conclusions should be extended, also taking into account practical and policy implications and directions for future research.
Best wishes!
Author Response
Comment: In my opinion, it is still missing a section on the literature review, that can be split from the introduction. Moreover, conclusions should be extended, also taking into account practical and policy implications and directions for future research.
Thank you for your suggestions. We have added the detailed literature review in p. 1 line 39 to p.2 line 79. Please note that the added texts were placed within the Introduction section in spite of the reviewer’s comment that such review could be split from introduction; this text placement was required to avoid the redundancy of texts and complexity of manuscript structure. We believe that this change is acceptable to remain the readability of the manuscript.
In regard to Conclusions, we have added the mention of policy implication to Discussion section (p.15, lines 464-469), and added the texts of practical and political implications, and future research directions to Conclusions section (p.16, lines 492-496), for consistency. We are convinced that these revisions contributed to the improvement of manuscript.